# Social and health system factors associated with maternal mortality in Eastern and Western China: Population health estimates using provincial-level data

Xiaojing Zeng[1,2], Dongjian Yang[1,2], Shiyang Li[3], Xiaolin Hua[1], Yanlin Wang[1,2], Jun Zhang[1,3]*, Zhiwei Liu[1,2]*

1 The International Peace Maternity and Child Health Hospital, Shanghai Jiao Tong University School of Medicine, Shanghai, China, 2 Shanghai Key Laboratory of Embryo Original Diseases, Shanghai, China, 3 Ministry of Education-Shanghai Key Laboratory of Children's Environmental Health, Xinhua Hospital, Shanghai Jiao Tong University School of Medicine, Shanghai, China

☺ These authors contributed equally to this work.
* junjimzhang@sjtu.edu.cn (JZ); liuzhiwei@hotmail.com (ZL)

## Abstract

### Background

Globally, maternal mortality is off track in achieving the Sustainable Development Goals by 2030. Over the past two decades, China has dramatically reduced maternal mortality in more developed (eastern) and less developed (western) regions. An understanding of the social and health system factors associated with maternal mortality in China may be helpful for countries attempting to meet the 2030 targets and beyond.

### Methods and findings

We analyzed provincial-level data on maternal mortality and social and health system factors from the National Health Statistics Yearbooks and China Statistical Yearbooks from 2004 to 2020. We investigated the factors associated with maternal mortality before and after 2013, the year that a historic national program, Reducing Maternal Mortality and Eliminating Neonatal Tetanus, came to an end. Bayesian kernel machine regression was employed to analyze social and health system factors (urbanization rate, per capita disposable income, average years of schooling, number of health technical personnel in maternal and child healthcare, number of hospital beds for obstetrics and gynecology, local fiscal expenditure on healthcare, prenatal booking rate, antenatal care rate, and hospital delivery rate) as a mixture and identify the factors with larger posterior inclusion probability and a higher value of the exposure–response relationship for the total and

which permits unrestricted use, distribution, and reproduction in any medium, provided the original author and source are credited.

**Data availability statement:** Data is publicly available at the National Health Commission website (http://www.nhc.gov.cn/mohwsbwst-jxxzx/tjzxtjcbw/tjsj_list.shtml), National Bureau of Statistics website (https://www.stats.gov.cn/sj/ndsj/), and Global Health Data Exchange (GHDx) online website (https://vizhub.healthda-ta.org/gbd-results/). All data inputs for BKMR are also provided in S16 Table.

**Funding:** This study was supported in part by the Shanghai Sailing Program (24YF2750300) (https://stcsm.sh.gov.cn) and the Postdoctoral Fellowship Program of China Postdoctoral Science Foundation (GZC20251538) (https://www.chinapostdoctor.org.cn/home) (X.Z.) and the Gates Foundation (INV-088623) (https://www.gatesfoundation.org) (J.Z.). The funders had no role in study design, data collection and analysis, decision to publish, or preparation of the manuscript.

**Competing interests:** The authors have declared that no competing interests exist.

**Abbreviation:** BKMR, Bayesian kernel machine regression; LMICs, lower-middle-income countries; MCH, maternal and child health; MDG, Millennium Development Goal; PCDI, per capita disposable income; PIP, posterior inclusion probability; SDG, Sustainable Development Goals; WHO, World Health Organization.

cause-specific maternal mortality. In the East, an increase in hospital delivery rate correlated with the decrease in total maternal mortality [posterior mean and standard deviation (SD): −14.8(1.5)] before 2013, and the urbanization rate was negatively associated with total maternal mortality [posterior mean and SD: −3.9(0.6)] after 2013. Hospital delivery, urbanization, local fiscal expenditure on healthcare, and antenatal care were the factors associated with reduced cause-specific maternal mortality in the East. In the West, an increase in antenatal care rate was associated with reduced total maternal mortality, with the posterior mean and SD of −33.8(6.8) and −11.5(4.1) before and after 2013, respectively. Hospital delivery and antenatal care were the factors associated with reduced cause-specific maternal mortality in the West. The main limitation of this study was the data constraints in the national statistics.

## Conclusions

Coverage of maternal care, health financing, and urbanization were the factors associated with the substantial reduction in maternal deaths in Eastern and Western China during 2004–2020. The improvement of the quantity and quality of antenatal care and hospital delivery may be a viable policy priority in less developed regions worldwide.

---

### Author summary
#### Why was this study done?

- Maternal mortality worldwide is not decreasing fast enough to reach global targets by 2030.

- China has made major progress in reducing maternal deaths in both wealthier and poorer regions over the past two decades.

- Learning from China's experience may help low- and lower-middle-income countries accelerate progress in reducing maternal mortality toward global goals.

#### What did the researchers do and find?

- We employed a Bayesian kernel machine regression model to identify the social and health system factors associated with decreased maternal mortality in China using nationwide provincial-level data from 2004 to 2020.

- Hospital delivery rate, antenatal care rate, urbanization rate, local fiscal expenditure on healthcare, and per capita disposable income were associated with improvements in maternal survival in both wealthier and poorer regions in China.

## What do these findings mean?

- To reduce maternal deaths in less developed regions, it may be vital to increase both the utilization and quality of antenatal care and facility birth, supported by adequate health funding and social improvements.

- The study's main limitation was the unavailability of more direct and comprehensive indicators of non-biomedical factors.

## Introduction

Tackling maternal mortality remains a complex task globally. Although some advances have been made, the progress has been slow and uneven. The declining trends in maternal mortality experienced a stagnation between 2016 and 2020 on a global level, and the global maternal mortality in 2020 was 223 per 100,000 livebirths [1]. The current state is off track in achieving the Sustainable Development Goals (SDG) target 3.1 of a maternal mortality below 70 per 100,000 livebirths by 2030 [1]. Of all maternal deaths, nearly 95% occurred in low- and lower-middle-income countries (LMICs), and 68 countries still had a maternal mortality of over 100 maternal deaths per 100,000 livebirths in 2020 [1]. Multisectoral action is required for less developed regions to transition from high to low maternal mortality stages. Evidence is needed on why some countries achieved continuous improvement and fulfilled the SDGs.

The World Health Organization (WHO) ranked China as one of the top 10 low- and middle-income fast-track countries in women's and children's health. It has achieved a considerable and consistent reduction in maternal mortality from 94.7 deaths in 1990 to 15.7 deaths per 100,000 livebirths in 2022 [2–4]. Currently, China has fulfilled the aim of reducing the maternal mortality to less than 70 per 100,000 livebirths in the SDGs. The more developed Eastern China has achieved Healthy China 2030's target level of 12 per 100,000 [2,3]. Although regional disparities also exist, and some western regions still have a long way to go, the gap has narrowed, with an 81% reduction in maternal mortality between 1996 and 2,018 in Western China [5]. These successes were not only attributable to the efforts aimed at addressing the leading biomedical causes of maternal mortality, but also the manifestations of the development in economic, political, and cultural systems [2]. The economic, political, and cultural superdeterminants may influence the social determinants of health and the performance of the healthcare system, which are the distal determinants of maternal health outcomes [6]. A comprehensive, quantitative analysis of the distal determinants contributing to the reduction of maternal mortality in the East and West may inform strategies for other less developed regions worldwide to achieve targets for 2030 and beyond.

Accordingly, guided by the recent review on superdeterminants of maternal mortality [6], we aimed to estimate the most contributing social and health system-related factors to the decline in total and cause-specific maternal mortality in China from 2004 to 2020. However, social and health system-related factors are often highly correlated, for example, between gross domestic product and life expectancy, and between population education levels and healthcare expenditure. Therefore, a new statistical method is required to address the collinearity of covariates. In recent years, Bayesian kernel machine regression (BKMR) has been increasingly used in environmental epidemiological studies since human beings are exposed to multiple chemicals simultaneously and these chemicals are often highly correlated [7,8]. BKMR is a non-parametric approach that combines Bayesian and statistical learning methods to estimate the joint exposure–response function iteratively using a Gaussian kernel function [7,8]. Under a hierarchical variable selection approach, BKMR addresses the possibility of collinearity by concurrently estimating both the importance of groups of highly correlated exposures or exposures from common sources, and the individual components within the group, making it more suitable for co-occurring exposure data than a traditional single-exposure model [8]. Here, we applied this method to social epidemiology as social and health system-related factors also concurrently impact pregnant women. We aimed to estimate the posterior inclusion probability (PIP) and the single-exposure risk measures of 9 social and health system factors, including urbanization rate, per capita disposable income (PCDI), average years of schooling, number of health

technical personnel in maternal and child health (MCH), number of hospital beds for obstetrics and gynecology (Ob/Gyn), local fiscal expenditure on healthcare, prenatal booking rate, antenatal care rate, and hospital delivery rate, and identify the factors with higher values of exposure–response relationships for reduced maternal mortality in China, thereby drawing implications for LMICs around the world.

## Methods

### Data source

Our study used publicly available databases, including the Global Burden of Diseases, Injuries, and Risk Factors Study 2021 (GBD 2021), the National Health Statistics Yearbooks, and the China Statistical Yearbooks. Guided by the framework and the list of non-biomedical determinants of maternal mortality proposed by Souza and colleagues [6], we selected the relevant social and health system-related factors from our data source (S1 Table). We restricted the analysis to the period between 2004 and 2020 because data on total and cause-specific maternal mortality at the provincial level are available during this period.

The National Health Statistical Yearbook is an annual statistical compendium documenting the progression of China's healthcare system and population health status [9]. Data on the maternal mortality, proportions of cause-specific maternal deaths, hospital delivery rate, antenatal care rate, prenatal booking rate, local fiscal expenditure on healthcare, number of health technical personnel in MCH care per 1,000 livebirths (hereafter referred to as number of MCH personnel), and number of hospital beds for Ob/Gyn per 1,000 livebirths (hereafter referred to as number of Ob/Gyn beds) in 31 provinces in mainland China from 2004 to 2020 were obtained from the National Health Statistics Yearbook (2005–2021). In the National Health Statistical Yearbook, the causes of maternal deaths were only classified as hemorrhage, coexisting medical conditions, hypertensive disorders in pregnancy (HDP), amniotic fluid embolism, and others. Due to a lack of detailed classification of causes of maternal deaths in the National Health Statistical Yearbook, we also extracted the data on the number and rate of deaths due to maternal disorders, including maternal hemorrhage, maternal sepsis and other maternal infections, maternal hypertensive disorders, obstructed labor and uterine rupture, abortion and miscarriage, ectopic pregnancy, indirect maternal deaths, late maternal deaths, maternal deaths aggravated by HIV/AIDS, and other direct maternal disorders, for females aged between 15 and 49 in China during 2004–2020 from GBD 2021 to identify the major causes of maternal mortality [10]. Indirect maternal deaths are due to existing diseases that are exacerbated by pregnancy. Coexisting medical conditions recorded in the National Health Statistical Yearbook specifically refer to internal medicine diseases that develop during pregnancy or pre-exist and are aggravated by pregnancy. The China Statistical Yearbook is an annual statistical publication that reflects China's economic and social development [11]. Data derived from the China Statistical Yearbook (2005–2021) include PCDI, urbanization rate, and the population by education attainment level for females in 31 provinces on the mainland of China from 2004 to 2020.

### Definition of variables

Maternal mortality refers to the number of maternal deaths per 100,000 livebirths [12]. The National Health Statistical Yearbook defines maternal deaths as any death of a woman while pregnant or within 42 days of the termination of the pregnancy from any cause related to the pregnancy or its management, but not from accidental or incidental causes [12]. Hospital delivery rate refers to the ratio of pregnant women delivered in hospitals to the number of livebirths. Antenatal care rate is the ratio of pregnant women who received at least one antenatal visit to the number of livebirths. Prenatal booking rate is the ratio of pregnant women who have maternal health records established by healthcare providers to the number of livebirths. Local fiscal expenditure on healthcare is the government's expenditure on medical and healthcare services, medical subsidies, health administration, and family planning. The number of MCH personnel is the number of professional staff engaged in MCH healthcare services per 1,000 livebirths. The number of Ob/Gyn beds refers to

the year-end number of beds in Ob/Gyn departments per 1,000 live births. Urbanization rate measures the proportion of a population residing in cities and towns. PCDI refers to the average income obtained by dividing the total disposable income by the resident population. To calculate the average years of schooling for females, the population by education attainment level for females was multiplied by the duration of the corresponding level first, and then the sum of these results was divided by the female population aged 6 and older.

## Statistical analysis

The eastern and western regions of China still have a disparity in socioeconomic development. We used the Aihui-Tengchong line as the geographical dividing line proposed by the Chinese geographer Huanyong Hu to divide China into eastern and western regions based on population density, natural conditions, and social, economic and human activities [13]. Maternal mortality in the East and West was calculated as the weighted average of maternal mortality in the 24 eastern provinces (Anhui, Beijing, Chongqing, Fujian, Guangdong, Guangxi, Guizhou, Hainan, Hebei, Heilongjiang, Henan, Hubei, Hunan, Jiangsu, Jiangxi, Jilin, Liaoning, Shandong, Shanghai, Shanxi, Sichuan, Tianjin, Yunnan, Zhejiang) and 7 western provinces (Gansu, Inner Mongolia, Ningxia, Qinghai, Shaanxi, Tibet, Xinjiang), respectively. The cause-specific mortality rate for each province was estimated by multiplying the maternal mortality and the proportion of maternal deaths from a specific cause.

The following data were missing for all provinces: local fiscal expenditure on healthcare between 2004 and 2006, PCDI in 2004, and the number of health technical personnel in maternal and child healthcare per 1,000 livebirths in 2009. For missing data imputation, we fitted a linear regression model for year and each of these three factors in each province. We used linear regression estimate imputation because the socio-economic missing value has a correlation with time and is at the beginning of the time [14]. We calculated the Spearman correlation coefficients among these social and health system factors (S1 Fig). Based on the correlation patterns and the sociologically-informed properties of these factors, we partitioned the 9 factors into 4 groups for the hierarchical variable selection in BKMR: 1) social determinants, including urbanization rate, PCDI, and average years of schooling; 2) health resources, including the number of MCH personnel and the number of Ob/Gyn beds; 3) health financing, including local fiscal expenditure on healthcare; and 4) coverage of maternal care, including prenatal booking rate, antenatal care rate, and hospital delivery rate.

The R package "bkmr" was used to perform the BKMR model before and after 2013, the year that a historic national program, Reducing Maternal Mortality and Eliminating Neonatal Tetanus, came to an end. In China, the annualized rate of reduction in maternal mortality was 4.9% from 1996 to 2000, which is slower than the Millennium Development Goal (MDG) 5 target of a 5.5% annual decline [15,16]. To meet the target pace in MDG 5, the Chinese government implemented the Reducing Maternal Mortality and Eliminating Neonatal Tetanus Program from 2000 to 2013 [2,17]. The program began in 378 counties across 12 western provinces, expanded to 2,288 counties in all central and western provinces by 2009, and was nationalized in 2009.

All the social and health system factors were standardized for the BKMR analysis. The kmbayes() function was used to fit the BKMR models with 10,000 Markov chain Monte Carlo iterations. Each province was assigned an identifier, so that a random intercept was included in the model to account for the correlation between repeated measures within the same province. The embedded hierarchical variable selection was performed to estimate the PIP of each of the 4 groups (group PIP, GroupPIP), and each factor within a group (conditional PIP, CondPIP). Both GroupPIP and CondPIP are indices that range from 0 to 1. A larger GroupPIP suggests that this group is relatively more likely to be associated with the outcome than other groups. Similarly, a larger ConPIP suggests that the factor is more likely to be associated with the outcome within the group.

To summarize the contribution of each single exposure to the response, we used the SingVarRiskSummaries() function to calculate the exposure–response relationship associated with a change in a single factor from its 75th percentile to its 25th percentile when other factors are fixed to a specific quantile. We used the following criteria to identify the

factors associated with decreased maternal mortality: 1) Among the four groups of social and health system factors, we first selected the groups with a GroupPIP ≥ 0.9; 2) For a group with a GroupPIP ≥ 0.9, the factor with the highest CondPIP was selected; 3) For the factor with the highest CondPIP within a group, significant and negative estimate of the single-exposure risk measures should be observed.

Several sensitivity analyses were performed to test the robustness of our results. 1) BKMR is less interpretable than traditional semi-parametric and parametric statistical models in providing a direct epidemiological interpretation with parsimonious parametric inference. Therefore, we further applied linear mixed-effects models to investigate the parameter inference for the factors identified by BKMR. The effect of each factor was estimated using univariate and multivariate linear mixed-effects models weighted by the number of livebirths in each province for each year. In the multivariate mixed-effects models, the other 8 social and health system factors were adjusted for when the effect of each factor was calculated. 2) Considering the potential impact of inflation on the analytical results, we conducted the BKMR analysis with government health expenditure adjusted for inflation. Government health expenditure was adjusted using 2012 as the base year for reference and the yearly inflation in the International Financial Statistics database [18]. 3) We used another method, multivariate imputation by chained equations (MICE) algorithm, to impute the missing values of exposures in 2004–2006 and 2009. Five imputed datasets were generated and modeled separately to re-examine the effects of these social and health system factors between 2004 and 2012. Results were pooled to obtain the final effect parameters. All the analyses were conducted in R 4.3.6. This study is reported as per the Guidelines for Accurate and Transparent Health Estimates Reporting (GATHER) statement (S1 Checklist).

## Results

### Trends in maternal mortality in Eastern and Western China from 2004 to 2020

In both the East and West, maternal mortality declined substantially, and the urban-rural gap in maternal mortality also narrowed between 2004 and 2020 (Fig 1). However, regional differences in maternal mortality persisted between the East and West during this period (S2 Fig). The characteristics of social and health system-related factors in the East and West before and after 2013 are shown in Table 1.

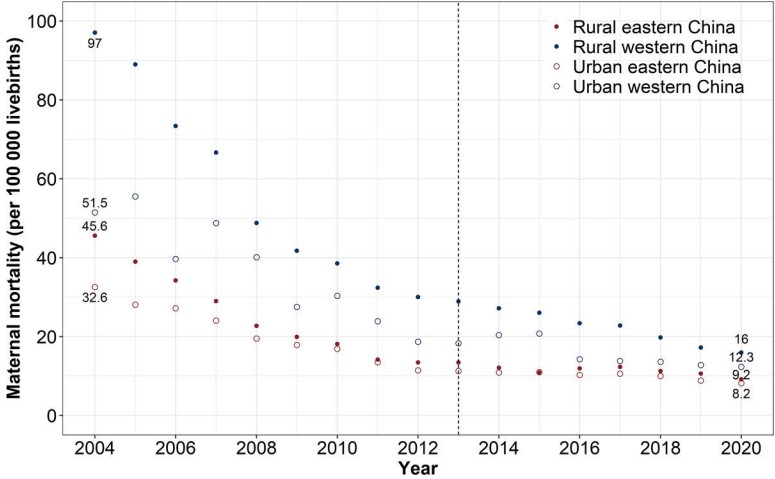

**Fig 1. Trends in maternal mortality in Eastern and Western China during 2004-2020.** The dashed line represents the year 2013 when the Reducing Maternal Mortality and Eliminating Neonatal Tetanus Program (2000–2013) ended.

**Table 1. Characteristics of social and health-system factors in Eastern and Western China during 2004-2012 and 2013–2020.**

| Factor | Eastern China | | Western China | |
|---|---|---|---|---|
| | 2004–2012 | 2013–2020 | 2004–2012 | 2013–2020 |
| Hospital delivery rate (%) | 98.7 [91.7, 99.7] | 100 [99.9, 100] | 89.3 [77.2, 97.4] | 99.7 [98.6, 100] |
| Antenatal care rate (%) | 92.9 [88.8, 96.5] | 97.2 [96.0, 98.3] | 92.4 [85.7, 95.9] | 97.1 [94.7, 97.7] |
| Prenatal booking rate (%) | 92.3 [85.5, 96.6] | 96.2 [94.2, 98.3] | 91.9 [84.3, 96.1] | 95.7 [92.7, 97.7] |
| Local fiscal expenditure on healthcare (100 million Chinese Yuan) | 147 [90, 219] | 444 [302, 617] | 54 [34, 116] | 203 [104, 313] |
| Urbanization rate (%) | 45.8 [35.9, 55.9] | 58.4 [52.7, 67.4] | 39.8 [29.5, 44.9] | 52.1 [46.3, 58.3] |
| PCDI (10 thousand Chinese Yuan) | 1.0 [0.7, 1.4] | 2.4 [1.9, 3.0] | 0.73 [0.55, 0.98] | 1.9 [1.5, 2.2] |
| Average years of schooling for females (years) | 8.0 [7.5, 8.9] | 8.8 [8.4, 9.2] | 7.7 [6.5, 8.2] | 8.5 [7.6, 9.0] |
| Number of Ob/Gyn beds per 1,000 livebirths | 21.1 [16.9, 34.6] | 30.7 [25.3, 40.0] | 23.3 [20.0, 27.8] | 34.9 [28.1, 42.1] |
| Number of MCH personnel per 1,000 livebirths | 13.9 [11.0, 18.9] | 21.4 [15.2, 29.8] | 12.9 [9.1, 22.1] | 19.4 [12.0, 26.7] |

Note: Data are median [Q1, Q3]. MCH, maternal and child health; Ob/Gyn, obstetrics and gynecology; PCDI, per capita disposable income.

Based on the data from the GBD 2021 study, maternal hemorrhage, indirect maternal deaths, other direct maternal disorders, and maternal hypertensive disorders remained the leading causes of maternal mortality in China from 2004 to 2020 (Fig 2 and S2 and S3 Tables). We further utilized provincial-level data from the National Health Statistics Yearbooks to estimate the cause-specific mortality rates in the East and West. Due to the lack of a detailed classification of maternal disorders in the provincial-level data comparable to that in the GBD study, we focused on hemorrhage, coexisting medical conditions, and HDP in the following analysis. These major causes of maternal mortality have been steadily declining in both the East and West from 2004 to 2020 (Fig 3).

## Factors associated with total and cause-specific maternal mortality in Eastern and Western China

Using BKMR, we identified five factors associated with the decreased maternal mortality based on the results of Group-PIP, CondPIP, and single-exposure risk measures (S4–S7 Tables and S3–S6 Figs): hospital delivery rate, antenatal care rate, urbanization rate, local fiscal expenditure on healthcare, and PCDI (Fig 4A–4D).

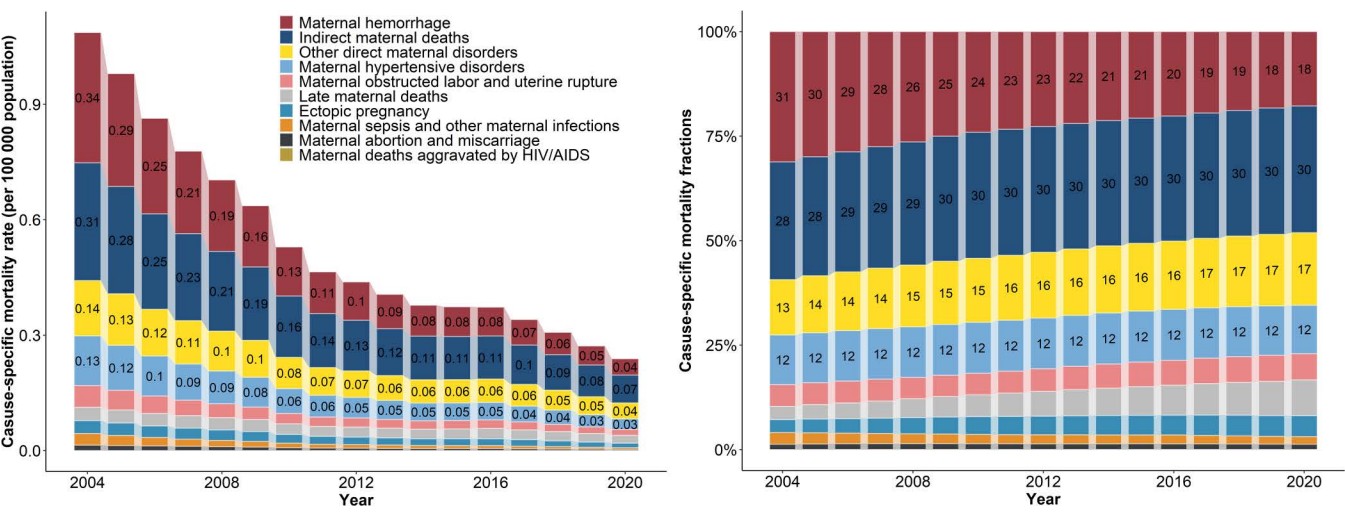

**Fig 2. Cause-specific maternal mortality rate (left panel) and fractions (right panel) in China, 2004–2020.** The top four causes are labeled with the mortality rate (left panel) and fraction (right panel).

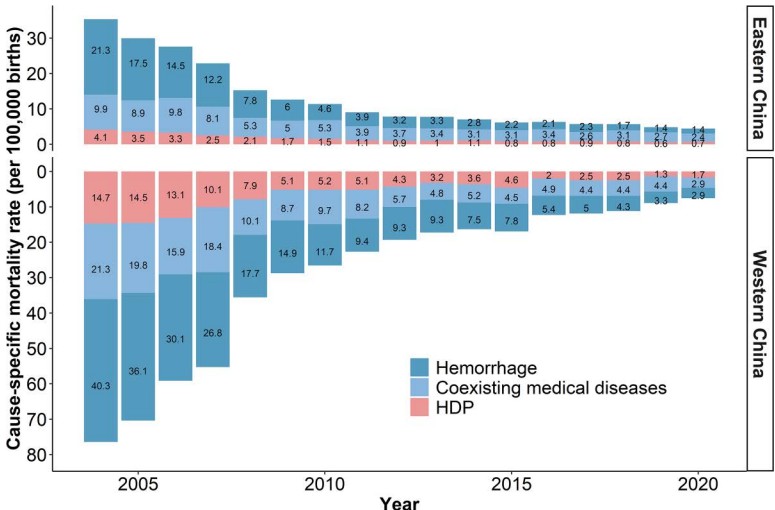

**Fig 3. Cause-specific maternal mortality rate for hemorrhage, coexisting medical diseases, and HDP in Eastern and Western China, 2004–2020.** HDP, hypertensive disorders in pregnancy.

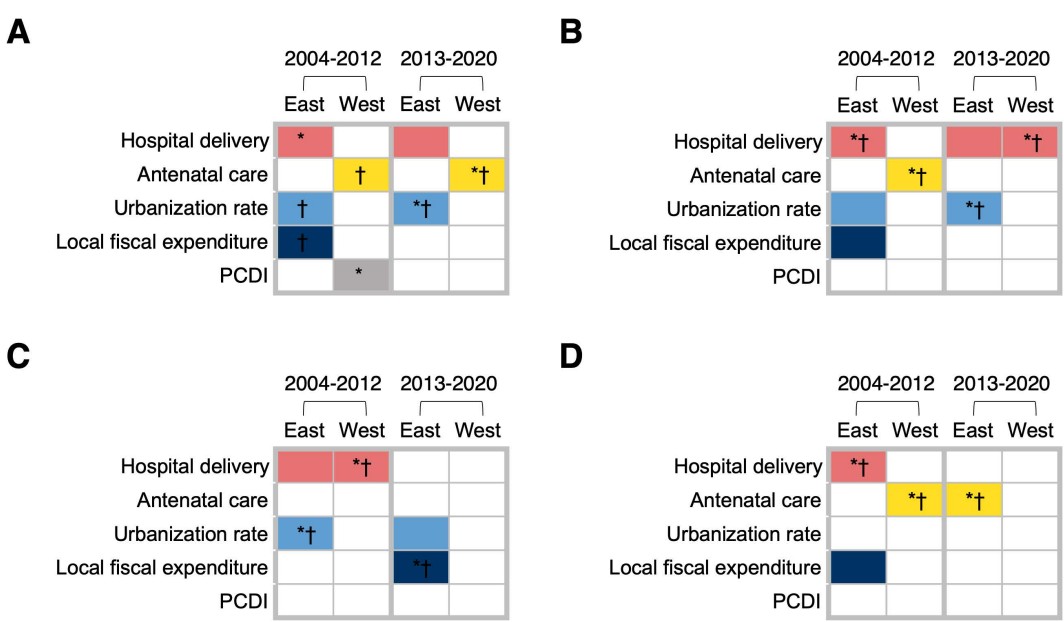

**Fig 4. Factors associated with decreased (A) total maternal mortality, (B) maternal mortality due to hemorrhage, (C) maternal mortality due to coexisting medical diseases, and (D) maternal mortality due to hypertensive disorders in pregnancy in Eastern and Western China during 2004–2012 and 2013–2020.** The asterisk * and dagger † symbols in the cell represent that the factor contributes the most to the exposure–response relationship when all other factors are fixed at their 25th and 75th percentiles, respectively. White cell means that the factor is not identified as the component associated with reduced maternal mortality in the mixture. PCDI, per capita disposable income.

For the reduction of total maternal mortality before 2013, the factors with larger GroupPIP and CondPIP, and significant single-exposure risk measures in the East were hospital delivery rate, urbanization rate, and local fiscal expenditure on healthcare (Figs 4A and S3 and S4 Table). When all other factors were at a lower level (25th percentile), an increase in hospital delivery rate from 91.7% to 99.7% was associated with a higher value of the exposure–response relationship with

a posterior mean and standard deviation (SD) of −14.8(1.5) (S3 Fig). When all other factors reached a high level (75th percentile), an increase in urbanization rate from 35.9% to 55.9% and an increase in local fiscal expenditure on healthcare from 9 billion to 21.9 billion Chinese Yuan were associated with larger responses in the decrease in maternal mortality [posterior mean and SD for urbanization rate: −5.8(2.6); posterior mean and SD for local fiscal expenditure on healthcare: −5.1(1.4)]. Before 2013, the factors identified by BKMR in the western region were PCDI and antenatal care rate (Figs 4A and S4 and S5 Tables). When other factors were at a lower level, an increase in PCDI from 5,500 to 9,800 Chinese Yuan [posterior mean and SD: −43.4 (8.1)] and an increase in antenatal care rate from 85.7% to 95.9% [posterior mean and SD: −33.8 (6.8)] were associated with larger single-exposure risk measures. An increase in antenatal care rate from 85.7% to 95.9% was also associated with a higher response in the decrease in maternal mortality when other factors were at a higher level [posterior mean and SD: −21.4(9.6)] (S4 Fig). After 2013, the urbanization rate and hospital delivery rate had higher GroupPIP and CondPIP, and significant single-exposure risk measures for the reduction of total maternal mortality in the East. An increase in the urbanization rate from 52.7% to 67.4% was associated with a larger response when considering the single-exposure risk measures, with a posterior mean and SD of −3.9(0.6) when other factors were at a lower lever (Figs 4A and S5 and S6 Table). In the West after 2013, an increase in antenatal care rate from 94.7% to 97.7% was associated with a larger response when considering the single-exposure risk measures, with a posterior mean and SD of −11.5(4.1) when other factors were at a lower level (Figs 4A and S6 and S7 Table).

The reduction of deaths due to hemorrhage was mainly correlated with hospital delivery rate, urbanization rate, and local fiscal expenditure on healthcare in the East during 2004–2012, with hospital delivery rate having a higher single-exposure risk measure, and hospital delivery rate and urbanization rate during 2013–2020, with urbanization rate having a higher single-exposure risk measure in the mixture of factors (Figs 4B, S3 and S5 and S4 and S6 Tables). In the West, however, antenatal care rate and hospital delivery rate were the factors with larger GroupPIP and CondPIP, and significant single-exposure risk measures before and after 2013, respectively (Figs 4B, S4 and S6 and S5 and S7 Tables).

For the reduction of maternal mortality caused by coexisting medical diseases in the East, the urbanization rate and local fiscal expenditure on healthcare were the factors associated with higher values of the exposure–response relationship before and after 2013, respectively (Figs 4C, S3 and S5 and S4 and S6 Tables). Before 2013, hospital delivery rate was also a factor associated with the decreased maternal mortality due to coexisting medical diseases in both the East and the West (Figs 4C and S3–S4 and S4–S5 Tables).

Hospital delivery rate and local fiscal expenditure on healthcare were the factors with larger GroupPIP and CondPIP, and significant single-exposure risk measures for the decline in deaths due to HDP in the East before 2013, and hospital delivery rate was associated with a larger single-exposure risk measure (Figs 4D and S3 and S4 Table). Antenatal care rate was the factor associated with a higher value of the exposure–response relationship in the West during 2004–2012, and in the East during 2013–2020 (Figs 4D, S4 and S5, and S5–S6 Tables).

### Sensitivity analyses

S8 and S9 Tables show the results of linear mixed-effects models for the factors identified by BKMR, including hospital delivery rate, antenatal care rate, urbanization rate, local fiscal expenditure on healthcare, and PCDI. Consistent with the results in the primary analyses, hospital delivery rate was negatively associated with total and cause-specific maternal mortality in the East before 2013. When adjusting for the other eight factors, each additional percentage of hospital delivery rate was associated with nearly one percent reduction in total maternal mortality and maternal death due to hemorrhage. In the West before 2013, each additional percentage of antenatal care rate was associated with 2% and 1% of reduction in maternal death due to hemorrhage and HDP, respectively, when adjusting for the other eight factors. The observations for the urbanization rate and PCDI also aligned with the primary results.

The results of BKMR analysis, with government health expenditure adjusted for inflation, are shown in S7–S11 Figs and S10–S13 Tables. The results of GroupPIP, CondPIP, and single-exposure risk measures did not change substantially.

When using datasets with imputations for missing data by MICE, the associations between factors and outcomes remained robust (S12–S14 Figs and S14–S15 Tables).

## Discussion

In this study, we conducted a mixture analysis to quantify the exposure–response relationships of social and health system-related factors and the decline of maternal mortality in Eastern and Western China from 2004 to 2020. We found that hospital delivery, antenatal care, urbanization, local fiscal expenditure on healthcare, and PCDI were the factors associated with the remarkable progress in maternal survival in both the eastern and western regions during this period. In addition, we observed the differential impact of different factors on cause-specific maternal mortality in the eastern and western regions before and after 2013, which may inform strategies for other less developed areas in the coming decades.

In our mixture analysis considering the simultaneous effects of multiple distal factors, the coverage of maternal care, including antenatal care and hospital delivery, was a key component associated with the decline in maternal deaths in China. As an indispensable part of the continuum of maternal healthcare, antenatal care reduces maternal mortality through timely detection and management for chronic conditions and pregnancy complications, particularly for individuals at increased risk of developing complications during labor and delivery [19]. A minimum of eight antenatal visits is recommended by WHO to reduce perinatal mortality [19]. In-hospital delivery reflects the accessibility of hospitals and the availability of skilled attendants [20]. With more skilled health workforce and more advanced equipment and infrastructure, hospital-based care can offer opportunities for early recognition of labor complications, facilitate timely treatment and emergency services, and provide better protection against mortality [21,22]. High-quality facility birth has been recommended globally as a strategy to reduce maternal mortality, especially in less developed regions where many women give birth at home with a non-certified birth attendant rather than in hospitals with professional health staff [23,24]. Using simulation modeling, Ward and colleagues found that for countries with maternal mortality levels between 140 and 1,000 per 100,000 livebirths, the most effective policy to reduce maternal mortality between 2030−2050 is increasing facility births, which is projected to prevent 22.6%−51.5% of maternal deaths [25].

Although the coverage of antenatal care and facility delivery increased in LMICs over the past several decades, 11% of women in 54 Countdown to 2030 priority countries had no antenatal care during pregnancy during 2012–2018, and 31 LMICs still had less than 80% utilization of facility delivery services in 2018 [26,27]. In countries with a maternal mortality of ≥700 per 100,000 livebirths, the coverage of at least one antenatal care visit and institutional birth was only 66% and 36% during 2000–2020, respectively [28]. Qualitative evidence in LMICs has found that the cost of care, distance to the facility, perceived quality of care, and social and cultural context are all barriers to accessing antenatal care and facility-based delivery [29–32].

It should also be noted that without ensuring adequate quality of care, increased coverage of maternal care does not necessarily improve maternal survival [33–36]. Increasing the proportion of women delivering in medical facilities is estimated to reduce total maternal death, death from hypertensive disorders and from hemorrhage by merely 29%, 29% and 53%, respectively [33]. Instead, an integrated strategy to simultaneously increase facility births, improve the availability of clinical services and quality of care at facilities, and improve linkages to care could substantially improve maternal health outcomes and reduce total maternal death, death from hypertensive disorders and from hemorrhage by 51%, 60% and 79%, respectively [33]. In addition to the recommendation on the number of antenatal visits, WHO also emphasizes the interventions to improve the quality of antenatal services in LMICs [19].

In China, a comprehensive strategy consisting of multidimensional interventions was employed in the Reducing Maternal Mortality and Eliminating Neonatal Tetanus Program to improve both the quantity and quality of maternal care and in-hospital delivery [2,17,37–39]. First, hospital delivery was subsidized by the government, and on average, women in rural areas who delivered in a hospital received a ¥500 subsidy directly at the time of discharge. Second, public awareness campaigns utilizing posters, cartoons, slogans, and other formats were organized, and primary healthcare staff and

village cadres worked together to provide health education and advocate for maternal care and hospital-based childbirth. Third, 24-hour obstetric emergency centers were built at the county level and pregnant women screened as high risk in the primary health system were referred to hospitals. The functions of traditional birth attendants were transformed from assisting home delivery to supporting antenatal care, referrals and the advocacy of hospital delivery. Fourth, equipment for emergency and routine obstetric care was provided to township health centers, and experts from provincial and municipal hospitals were dispatched to county and township hospitals to train local health staff. In addition to this program, a series of regulations and services were also instituted to improve the quality of antenatal care. The Ministry of Health in China required a minimum of five antenatal visits, including one visit in the first trimester, two in the second trimester, and two in the third trimester [40]. In 2009, the Chinese government launched the National Essential Public Health Services Program to provide universal coverage of essential health services nationwide, including five antenatal examinations free of charge at the point of care [17]. The cumulative effect of these programs and services resulted in a 68.7% increase in the hospital delivery rate in the project area, and the coverage of at least five antenatal visits increased from 38% in 2003 to 57% in 2,011 in Western China [2,9,17]. A subnational analysis revealed that the annualized rate of decline in maternal mortality improved year-over-year after the program's implementation and reached a mean county-level rate of 9.3% during 2005–2015 [15]. Maternal mortality in the project area decreased by 67.1% from 2000 to 2013 [2]. The effect of hospital delivery and antenatal care on maternal survival has persisted in recent decades, as shown in our mixture analysis after 2,013 in both the East and West.

Using a mixture analysis, we found that the effects of hospital delivery and antenatal care on maternal survival were paralleled by the contribution of social determinants and health financing. Social determinants, such as income levels, can affect maternal health through changes in women's decisions to seek care. Women living in urban areas have more access to high-quality and high-acuity healthcare during pregnancy and childbirth [41–43]. Financing is a pivotal factor contributing to universal health coverage, and allocating adequate resources and effective healthcare financing is one of the cross-cutting actions proposed by WHO for ending preventable maternal deaths [2,44]. Unfortunately, recent reports published by the World Bank Group revealed a steady decline in per capita government health spending from 2020 to 2023 and troubling trends in the prioritization of health across LMICs, and the expected spending trajectories in LMICs remain below the minimum needed to meet the SDGs by 2030 [45,46]. These financial shortfalls underscore the urgent need to increase government commitment to health and reform financing policies [45,46].

For LMICs with a high burden of maternal mortality, lessons could be learnt from China's remarkable success. Here, we identified the factors associated with higher values of the exposure–response relationship in order to suggest the priorities that can be considered in policy and program planning. This is particularly relevant for resource-limited settings, where simultaneous optimization of multiple factors may not be feasible. The improvement of the quantity and quality of facility births and antenatal care might be the leverage point to accelerate the decline in maternal mortality. In China, antenatal attendance and safe birth in health facilities were underpinned by strong government commitment to maternal health and a series of long-term strategies, featuring in the subsidies and health education to reduce the financial and cultural barriers that delay the decision to seek care, and the green channel service for obstetric emergencies and the professionalization of obstetric healthcare to reduce the delays in reaching a health facility and receiving adequate care [17,47]. Additionally, a hierarchical healthcare network spanning from national to village levels might be considered in LMICs to establish a well-functioning referral system and foster close working partnerships between hospitals and primary healthcare facilities.

A major strength of our study is the quantification of the impact of non-medical distal factors associated with maternal survival in the East and West over a nearly 20-year timescale. We included a list of socioeconomic, financial and system-related factors at the national and subnational levels. These factors are closely linked to the policies and strategies China has adopted over the past few decades, from which lessons can be extracted to share with less developed regions in the world. In addition to total maternal mortality, we also measured the distal factors for reduced maternal deaths due to major biomedical causes to provide insights to guide targeted intervention programs for each maternal disorder. Furthermore,

we utilized mixture analysis to simultaneously estimate the effects of these co-occurring factors, which can address the issue of collinearity and rank the values of exposure–response relationships for these factors.

Our analysis is limited by the unavailability of some direct indicators of non-medical factors. For example, although urbanization may represent the development of healthcare services and transportation, it is not a direct measure of the distance to a health facility. However, we used multiple indicators in addition to the urbanization rate, including the number of medical personnel and hospital beds, as well as the rate for prenatal booking, antenatal care and hospital delivery, to reflect the accessibility of health services. Besides, data on other non-medical factors are unavailable, such as gender and social bias against women. A global analysis including more distal factors is warranted to rank the values of exposure–response relationships for these co-occurring contributors to decreased maternal mortality. Second, data at the provincial level outside the 2004–2020 range are unavailable in the current study. However, due to the rapid shifts in economic, political and cultural contexts, the social and system factors before the MDG era might have little applicability in the following decades. Besides, according to Souza and colleagues' obstetric transition model [6], China is now in the category of Stage 4B, which represents a very low maternal mortality (maternal mortality <20). Therefore, the experience in China after 2020 would be impractical for countries with a high maternal mortality burden. Third, data on rural and urban areas at the provincial level are unavailable for most of the factors. Therefore, we are not able to analyze the contributing factors by rural and urban areas in the East and West. Fourth, both too much, too soon and too little, too late are associated with maternal mortality and have been found to coexist in China and other LMICs [48]. However, due to the lack of data reflecting over-medicalization, we were unable to examine the impact of insufficient and excessive medicalization on maternal mortality in China in the past decades. Fifth, due to the data constraints in the National Health Statistics Yearbooks, our analysis only provides the data for maternal deaths from coexisting medical diseases rather than the indirect causes, including both previously existing conditions and non-communicable diseases. It is of great importance to conduct further investigations of the distal factors associated with indirect maternal deaths, which remain the second largest cause of maternal deaths globally [49]. Finally, this study relied on national annual statistics, which may be subject to reporting biases and regional inconsistencies in data collection and data quality. This limitation should be considered when interpreting the results.

Globally, maternal mortality is currently not on track to meet the SDG target, with a disproportionate burden in LMICs [1]. Although the social and health system factors are all crucial to maternal mortality, investigating the values of exposure–response relationships for these factors can provide practical guidance for policy and program planning in these developed regions. For other less developed regions around the world, enhancing both the quantity and quality of antenatal care and facility births may be considered a policy priority. Health financing and social determinants are also crucial in ensuring universal coverage of maternal health services in these regions. Actions targeting these non-biomedical, distal factors are necessary to hold any hope of achieving the SDG target 3.1.

## Supporting information

**S1 Table. Social and health system-related factors of maternal mortality.** Note: Ob/Gyn, obstetrics and gynecology. (DOCX)

**S2 Table. Cause-specific maternal mortality rate in China, 2004–2020.** (DOCX)

**S3 Table. Cause-specific maternal mortality fractions in China, 2004–2020.** (DOCX)

**S4 Table. Group and conditional posterior inclusion probabilities for each factor in Eastern China, 2004–2012, using Bayesian Kernel Machine Regression hierarchical variable selection.** Note: GroupPIP, group posterior

inclusion probabilities; CondPIP, conditional posterior inclusion probabilities; MCH, maternal and child health; Ob/Gyn, obstetrics and gynecology; PCDI, per capita disposable income.
(DOCX)

**S5 Table. Group and conditional posterior inclusion probabilities for each factor in Western China, 2004–2012, using Bayesian Kernel Machine Regression hierarchical variable selection.** Note: GroupPIP, group posterior inclusion probabilities; CondPIP, conditional posterior inclusion probabilities; MCH, maternal and child health; Ob/Gyn, obstetrics and gynecology; PCDI, per capita disposable income.
(DOCX)

**S6 Table. Group and conditional posterior inclusion probabilities for each factor in Eastern China, 2013–2020, using Bayesian Kernel Machine Regression hierarchical variable selection.** Note: GroupPIP, group posterior inclusion probabilities; CondPIP, conditional posterior inclusion probabilities; MCH, maternal and child health; Ob/Gyn, obstetrics and gynecology; PCDI, per capita disposable income.
(DOCX)

**S7 Table. Group and conditional posterior inclusion probabilities for each factor in Western China, 2013——2020, using Bayesian Kernel Machine Regression hierarchical variable selection.** Note: GroupPIP, group posterior inclusion probabilities; CondPIP, conditional posterior inclusion probabilities; MCH, maternal and child health; Ob/Gyn, obstetrics and gynecology; PCDI, per capita disposable income.
(DOCX)

**S8 Table. Effects of factors on maternal mortality, using univariate linear mixed-effects models weighted by the number of livebirths.** Note: Effects are shown as estimates and the 95% confidence intervals in the linear mixed effects models.
(DOCX)

**S9 Table. Effects of factors on maternal mortality, using multivariate linear mixed-effects models weighted by the number of livebirths.** Note: Effects are shown as estimates and the 95% confidence intervals in the linear mixed effects model, adjusting for the other eight factors. For example, the effect of hospital delivery rate was estimated when urbanization rate, per capita disposable income, average years of schooling, number of health technical personnel in maternal and child health, number of hospital beds for obstetrics and gynecology, local fiscal expenditure on healthcare, prenatal booking rate, and antenatal care rate were adjusted for.
(DOCX)

**S10 Table. Group and conditional posterior inclusion probabilities for each factor in Eastern China, 2004–2012, using Bayesian Kernel Machine Regression hierarchical variable selection with fiscal expenditure adjusted for inflation.** Note: GroupPIP, group posterior inclusion probabilities; CondPIP, conditional posterior inclusion probabilities; MCH, maternal and child health; Ob/Gyn, obstetrics and gynecology; PCDI, per capita disposable income.
(DOCX)

**S11 Table. Group and conditional posterior inclusion probabilities for each factor in Western China, 2004–2012, using Bayesian Kernel Machine Regression hierarchical variable selection with fiscal expenditure adjusted for inflation.** Note: GroupPIP, group posterior inclusion probabilities; CondPIP, conditional posterior inclusion probabilities; MCH, maternal and child health; Ob/Gyn, obstetrics and gynecology; PCDI, per capita disposable income.
(DOCX)

**S12 Table. Group and conditional posterior inclusion probabilities for each factor in Eastern China, 2013–2020, using Bayesian Kernel Machine Regression hierarchical variable selection with fiscal expenditure adjusted for**

**inflation.** Note: GroupPIP, group posterior inclusion probabilities; CondPIP, conditional posterior inclusion probabilities; MCH, maternal and child health; Ob/Gyn, obstetrics and gynecology; PCDI, per capita disposable income.
(DOCX)

**S13 Table. Group and conditional posterior inclusion probabilities for each factor in Western China, 2013–2020, using Bayesian Kernel Machine Regression hierarchical variable selection with fiscal expenditure adjusted for inflation.** Note: GroupPIP, group posterior inclusion probabilities; CondPIP, conditional posterior inclusion probabilities; MCH, maternal and child health; Ob/Gyn, obstetrics and gynecology; PCDI, per capita disposable income.
(DOCX)

**S14 Table. Group and conditional posterior inclusion probabilities for each factor in Eastern China, 2004–2012, using Bayesian Kernel Machine Regression hierarchical variable selection with missing data imputed by MICE.** Note: GroupPIP, group posterior inclusion probabilities; CondPIP, conditional posterior inclusion probabilities; MCH, maternal and child health; Ob/Gyn, obstetrics and gynecology; PCDI, per capita disposable income.
(DOCX)

**S15 Table. Group and conditional posterior inclusion probabilities for each factor in Western China, 2004–2012, using Bayesian Kernel Machine Regression hierarchical variable selection with missing data imputed by MICE.** Note: GroupPIP, group posterior inclusion probabilities; CondPIP, conditional posterior inclusion probabilities; MCH, maternal and child health; Ob/Gyn, obstetrics and gynecology; PCDI, per capita disposable income.
(DOCX)

**S16 Table. Data inputs for BKMR.**
(XLSX)

**S1 Fig. Spearman correlation coefficient matrix for the non-medical factors.** Note: The number in the cells represent the Spearman correlation coefficient. The circles are filled clockwise as pie charts to indicate the positive values of the correlation coefficients, and the color intensity represents the strength or the correlation. The asterisks in the circles indicate the significant levels with * representing a *P*-value of < 0.05, ** representing a *P*-value of < 0.01, and *** representing a *P*-value of < 0.001. Ob/Gyn beds: Number of hospital beds for obstetrics and gynecology per 1,000 livebirths; MCH personnel: Number of health technical personnel in maternal and child healthcare per 1,000 livebirths; Years of schooling: Average years of schooling for females; PCDI: per capita disposable income; Hospital delivery: Hospital delivery rate; Local fiscal expenditure: Local fiscal expenditure on healthcare; Antenatal care: Antenatal care rate; Prenatal booking: Prenatal booking rate
(TIFF)

**S2 Fig. Map of the provincial-level maternal mortality in mainland China in 2004, 2011 and 2020.** Note: The source of the basemap is: https://cloudcenter.tianditu.gov.cn/administrativeDivision.
(TIFF)

**S3 Fig. Single-exposure risk measures for each factor in Eastern China, 2004–2012, using Bayesian Kernel Machine Regression hierarchical variable selection.** Note: The plot compares the exposure–response relationships associated with a change in a single exposure from the 75th percentile to the 25th percentile, when the other exposures are fixed at their 25th, 50th, and 75th percentiles. MCH, maternal and child health; Ob/Gyn, obstetrics and gynecology; PCDI, per capita disposable income; HDP, hypertensive disorders in pregnancy; q.fixed, quantiles at which to fix the remaining exposures.
(TIFF)

**S4 Fig. Single-exposure risk measures for each factor in Western China, 2004–2012, using Bayesian Kernel Machine Regression hierarchical variable selection.** Note: The plot compares the exposure–response relationships

associated with a change in a single exposure from the 75th percentile to the 25th percentile, when the other exposures are fixed at their 25th, 50th, and 75th percentiles. MCH, maternal and child health; Ob/Gyn, obstetrics and gynecology; PCDI, per capita disposable income; HDP, hypertensive disorders in pregnancy; q.fixed, quantiles at which to fix the remaining exposures.
(TIFF)

**S5 Fig. Single-exposure risk measures for each factor in Eastern China, 2013–2020, using Bayesian Kernel Machine Regression hierarchical variable selection.** Note: The plot compares the exposure–response relationships associated with a change in a single exposure from the 75th percentile to the 25th percentile, when the other exposures are fixed at their 25th, 50th, and 75th percentiles. MCH, maternal and child health; Ob/Gyn, obstetrics and gynecology; PCDI, per capita disposable income; HDP, hypertensive disorders in pregnancy; q.fixed, quantiles at which to fix the remaining exposures.
(TIFF)

**S6 Fig. Single-exposure risk measures for each factor in Western China, 2013–2020, using Bayesian Kernel Machine Regression hierarchical variable selection.** Note: The plot compares the exposure–response relationships associated with a change in a single exposure from the 75th percentile to the 25th percentile, when the other exposures are fixed at their 25th, 50th, and 75th percentiles. MCH, maternal and child health; Ob/Gyn, obstetrics and gynecology; PCDI, per capita disposable income; HDP, hypertensive disorders in pregnancy; q.fixed, quantiles at which to fix the remaining exposures.
(TIFF)

**S7 Fig. Factors associated with decreased total and cause-specific maternal mortality in Eastern and Western China during 2004–2012 and 2013–2020, using Bayesian Kernel Machine Regression hierarchical variable selection with fiscal expenditure adjusted for inflation.** Note: The asterisk * and dagger † symbols in the cell represent that the factor contributes the most to the exposure–response relationship when all other factors are fixed at their 25th and 75th percentiles, respectively. White cell means that the factor is not identified as the component associated with reduced maternal mortality in the mixture. CMD, coexisting medical diseases; HDP, hypertensive disorders in pregnancy; PCDI, per capita disposable income.
(TIFF)

**S8 Fig. Single-exposure risk measures for each factor in Eastern China, 2004–2012, using Bayesian Kernel Machine Regression hierarchical variable selection with fiscal expenditure adjusted for inflation.** Note: The plot compares the exposure–response relationships associated with a change in a single exposure from the 75th percentile to the 25th percentile, when the other exposures are fixed at their 25th, 50th, and 75th percentiles. MCH, maternal and child health; Ob/Gyn, obstetrics and gynecology; PCDI, per capita disposable income; HDP, hypertensive disorders in pregnancy; q.fixed, quantiles at which to fix the remaining exposures.
(TIFF)

**S9 Fig. Single-exposure risk measures for each factor in Western China, 2004–2012, using Bayesian Kernel Machine Regression hierarchical variable selection with fiscal expenditure adjusted for inflation.** Note: The plot compares the exposure–response relationships associated with a change in a single exposure from the 75th percentile to the 25th percentile, when the other exposures are fixed at their 25th, 50th, and 75th percentiles. MCH, maternal and child health; Ob/Gyn, obstetrics and gynecology; PCDI, per capita disposable income; HDP, hypertensive disorders in pregnancy; q.fixed, quantiles at which to fix the remaining exposures.
(TIFF)

**S10 Fig. Single-exposure risk measures for each factor in Eastern China, 2013–2020, using Bayesian Kernel Machine Regression hierarchical variable selection with fiscal expenditure adjusted for inflation.** Note: The plot

compares the exposure–response relationships associated with a change in a single exposure from the 75th percentile to the 25th percentile, when the other exposures are fixed at their 25th, 50th, and 75th percentiles. MCH, maternal and child health; Ob/Gyn, obstetrics and gynecology; PCDI, per capita disposable income; HDP, hypertensive disorders in pregnancy; q.fixed, quantiles at which to fix the remaining exposures
(TIFF)

**S11 Fig. Single-exposure risk measures for each factor in Western China, 2013–2020, using Bayesian Kernel Machine Regression hierarchical variable selection with fiscal expenditure adjusted for inflation.** Note: The plot compares the exposure–response relationships associated with a change in a single exposure from the 75th percentile to the 25th percentile, when the other exposures are fixed at their 25th, 50th, and 75th percentiles. MCH, maternal and child health; Ob/Gyn, obstetrics and gynecology; PCDI, per capita disposable income; HDP, hypertensive disorders in pregnancy; q.fixed, quantiles at which to fix the remaining exposures.
(TIFF)

**S12 Fig. Factors associated with decreased total and cause-specific maternal mortality in Eastern and Western China during 2004–2012, using Bayesian Kernel Machine Regression hierarchical variable selection with missing data imputed by MICE.** Note: The asterisk * and dagger † symbols in the cell represent that the factor contributes the most to the exposure–response relationship when all other factors are fixed at their 25th and 75th percentiles, respectively. White cell means that the factor is not identified as the component associated with reduced maternal mortality in the mixture. CMD, coexisting medical diseases; HDP, hypertensive disorders in pregnancy; PCDI, per capita.
(TIFF)

**S13 Fig. Single-exposure risk measures for each factor in Eastern China, 2004–2012, using Bayesian Kernel Machine Regression hierarchical variable selection with missing data imputed by MICE.** Note: The plot compares the exposure–response relationships associated with a change in a single exposure from the 75th percentile to the 25th percentile, when the other exposures are fixed at their 25th, 50th, and 75th percentiles. MCH, maternal and child health; Ob/Gyn, obstetrics and gynecology; PCDI, per capita disposable income; HDP, hypertensive disorders in pregnancy; q.fixed, quantiles at which to fix the remaining exposures.
(TIFF)

**S14 Fig. Single-exposure risk measures for each factor in Western China, 2004–2012, using Bayesian Kernel Machine Regression hierarchical variable selection with missing data imputed by MICE.** Note: The plot compares the exposure–response relationships associated with a change in a single exposure from the 75th percentile to the 25th percentile, when the other exposures are fixed at their 25th, 50th, and 75th percentiles. MCH, maternal and child health; Ob/Gyn, obstetrics and gynecology; PCDI, per capita disposable income; HDP, hypertensive disorders in pregnancy; q.fixed, quantiles at which to fix the remaining exposures.
(TIFF)

**S1 Checklist. Checklist of information that should be included in new reports of global health estimates.**
(DOCX)

## Author contributions

**Conceptualization:** Jun Zhang, Zhiwei Liu.

**Data curation:** Xiaojing Zeng, Dongjian Yang.

**Formal analysis:** Xiaojing Zeng, Dongjian Yang.

**Funding acquisition:** Xiaojing Zeng, Jun Zhang.

**Visualization:** Xiaojing Zeng, Dongjian Yang.

**Writing – original draft:** Xiaojing Zeng.

**Writing – review & editing:** Shiyang Li, Xiaolin Hua, Yanlin Wang, Jun Zhang, Zhiwei Liu.

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
