## [Editor Report · Decision Letter 0]

6 Aug 2025

Dear Dr Zhang,

Thank you for submitting your manuscript entitled "Social and Health System Determinants of Maternal Mortality in More and Less Developed Regions of China: Implications for SDG 2030" for consideration by PLOS Medicine.

Your manuscript has now been evaluated by the PLOS Medicine editorial staff as well as by an academic editor with relevant expertise and I am writing to let you know that we would like to send your submission out for external peer review.

For clinical studies, please upload a copy of your trial study protocol as a supporting information file. The study protocol should be the version submitted for approval to the institutional review board or ethics committee, should include any amendments to the study protocol, as well as the date of their approval by the institutional review or ethics committee. Please also detail any deviations from the study protocol in the Methods section of your manuscript. The editors will consider the protocol and study conduct prior to a final decision for external review.

Please re-submit your manuscript within two working days, i.e. by Aug 08 2025 11:59PM.

Kind regards,

Heather Van Epps, PhD

Consulting Editor

PLOS Medicine

---

## [Decision Letter · Decision Letter 1]

12 Sep 2025

Dear Dr Zhang,

Many thanks for submitting your manuscript "Social and Health System Determinants of Maternal Mortality in More and Less Developed Regions of China: Implications for SDG 2030" (PMEDICINE-D-25-02747R1) to PLOS Medicine. The paper has been reviewed by subject experts and a statistician; their comments are included below and can also be accessed here: [LINK]

As you will see, the reviewers find the work interesting and timely and have provide suggestions to improve the analyses, presentation and potential impact. After discussing the paper with the editorial team and an academic editor with relevant expertise, I'm pleased to invite you to revise the paper in response to the reviewers' comments. We plan to send the revised paper to some or all of the original reviewers, and we cannot provide any guarantees at this stage regarding publication.

We ask that you submit your revision by Sep 29 2025 11:59PM. However, if this deadline is not feasible, please contact me by email, and we can discuss a suitable alternative.

Don't hesitate to contact me directly with any questions.

Best regards,

Alison

Alison Farrell, PhD

Senior Editor

PLOS Medicine

afarrell@plos.org

Comments from the reviewers:

Reviewer #1: This article uses Bayesian kernel machine regression to perform variable selection on various provincial-level social determinants of health to examine associations with longitudinal maternal mortality markers. In general, this was a very well-written article, with strong visualizations, and utilizing a statistical methodology that has been used before in similar settings. I appreciated the clarity in which the variables were defined, and the background provided on China's impressive improvements in maternal mortality and public health were helpful and provided great context for the study. However, there is potential room for improvement, as outlined in the suggestions below. I am not necessarily asking for entire reanalyses of the data or even that the suggestions be run, but would like to engage with the authors and understand more regarding the following points.

I have two main comments: the first being that I didn't think the GP model was particularly necessary (in that use of this model and the interpretations that came out of it might be obfuscating simpler relationships), and the second being that some of the most relevant epidemiological interpretations weren't actually provided in terms of direct interpretations of exposure/response relatonships. There were only nine predictors considered in the study, with "exposure group 2" comprising only a single variable! I'm curious what a model that simply used all predictors or one with sociologically-informed variable selection with groups would look like; my guess is pretty similar in terms of coefficient estimates to the one presented after the complex statistical machinery.

As mentioned, another potential area for improvement would be in interpretability of the model. Although it is perhaps of interest to know that an increase in hospital relivery rate from Q1 to Q3 "contributed the most to the exposure-response relationship," this provides only part of the picture. Is it possible that the antenatal care rate going from Q1 to Q3 also "contributes" quite a great deal to this relationship? What are the estimated posterior means and 95% credible sets there? Additionally, it would have been helpful to have some sort of real-world idea of what going from Q1 to Q3 entailed for this variable in the text instead of having to look it up in Table 1.

On this note, the real-world implications of Q1 to Q3 changes also vary quite a bit. For instance, in Western China from 2004-2012, Q1-Q3 was 77.2 - 97.4 for hospital delivery rate, whereas in Eastern China from 2013-2020 that same Q1-Q3 was 99.9-100. To me, this seems to drive the story - there simply wasn't any meaningful variability in Eastern China in that time period, thus driving the storyline that hospital delivery "wasn't important" in explaining exposure/outcome relationships for that model. Would it not simply be more relevant to present results from a simpler model? Even a straightforward mixed-effects longitudinal model would be able to quantify something more directly relevant to public health/sociological understanding, something along the lines of "for this region in this time period, adjusting for xyz, each additional percentage of hospital delivery % was associated with an ##% reduction in maternal death due to hemorrhage" or something, instead of identifying the "most important variable" when going to Q1-Q3, knowing that this difference might be quite different across variables or even irrelevant in other cases (cf. the example earlier in this paragraph).

An observation regarding the modeling procedure is that fitting the Eastern/Western separately removes ability to borrow information (which the Bayesian paradigm is great for) across provinces in the other region, essentially fitting interactions across all variables. This is fine if this was the intention, but I also wonder what "one big model" with all provinces and regions would have looked like together, perhaps with terms/interactions corresponding to region. As a personal curiosity, I also wonder what an analysis weighted by population would look like - as is, Qinghai has the same weight as Sichuan in the model, though on absolute public health terms regarding mortality, it seems that we might care much more about Sichuan or that its contribution should be weighted much more. It may not be worth running the model, but a sensitivity analysis (perhaps of a simple mixed-effects model) may be interesting from a scientific point of view.

Finally, use of words like "attributable" or "most important" may unintentionally imply a causal relationship with these social determinants of health and reduced maternal mortality despite these being retrospective observational data. I would strongly encourage you to use more associational language in order not to inadvertently mislead less savvy readers.

As minor suggestions, I would have appreciated a bit more information regarding the imputation approach for missing determinants of health (what specific model was used? How much data was missing? Why not an imputation method that allows for valid inference, e.g., MICE, instead of deterministic imputation according to a model, thus under-representing variability both in terms of the variable itself but also through the fact that imputation had to be performed?). Furthermore, the term "posterior SD" does not seem to apply (e.g., in line 259) - my guess is that you are presenting 95% HPD credible intervals for the posterior mean. The term "interquartile range" is used incorrectly; the IQR specifically refers to the difference between Q1 and Q3, and so I recommend simply reporting [Q1 - Q3] instead. It does not appear that fiscal expenditure is adjusted for inflation - the nominal values reported mean quite different things, and perhaps would look less stark if adjusted to, say, 2010 RMB. Finally, the text in the figures were quite hard to read - please ensure that they are available at sufficient DPI in the final submission.

Reviewer #2: Manuscript title: Social and Health System Determinants of Maternal Mortality in More and Less Developed Regions of China: Implications for SDG 2030

Journal: PLOS Medicine

Summary

This manuscript analyzes maternal mortality in China from 2004-2020, using provincial-level data and Bayesian kernel machine regression (BKMR) to evaluate the relative contributions of social and health system determinants. The authors highlight hospital delivery, antenatal care, health financing, income, and urbanization as key factors behind China's remarkable reduction in maternal mortality, and frame lessons for low- and middle-income countries (LMICs) still striving toward SDG target 3.1.

The topic is timely and highly relevant for global health policy. The innovative application of BKMR to maternal health research is a clear methodological strength, and the East/West and pre-/post-2013 comparisons are well thought out. However, the manuscript would benefit from sharpening its narrative focus, simplifying presentation of results, and expanding the discussion of policy implications — particularly around quality of care. In addition, the quality of data collected from the national statistics especial in the 2010s may be problematic.

Major Comments

The manuscript has put overemphasis on descriptive results, which have been widely reported over the past decade. For example, Figures 1-3 and much of the Results reiterate well-documented declines in maternal mortality. These sections could be streamlined to allow more focus on the determinants analysis, which is the paper's main value-added.

The study used many data from the national annual statistics books published by the Chinese central and local governments. However, we all know that the quality of such routine data has been problematic. For example, China has got a huge population of rural-to-urban migrant workers (over 200 million, based on some reports). The health statistics in one province may not include those migrant populations who do not have official resident status. Such a practice was very common in the early 2000s in China. I am wondering how the authors addressed such a challenge. In addition, different provinces/regions may have different working definition of "urbanization rate". Is the indicator only the use of official residence status? Or the calculation has considered the rural-to-urban population living in cities, but registered in the rural areas?

It should also give clearer interpretation of BKMR Findings: while GroupPIP and CondPIP are presented, their practical meaning is not sufficiently explained. For a broad PLOS Medicine readership, please clarify: what does a CondPIP of 0.9 mean in policy terms? How large are these effects compared to known biomedical interventions?

The discussion highlights antenatal care and facility births, but the issue of quality of care is underdeveloped. WHO has emphasized that coverage alone is insufficient; over-medicalization (e.g., rising cesarean section rates in China) and quality deficits are important considerations. A more balanced discussion would strengthen the global relevance.

Minor Comments

Abstract: Too dense. Simplify by foregrounding the five main determinants identified (hospital delivery, antenatal care, financing, urbanization, income) and reduce descriptive statistics.

Figures: Figure 4 is central but overloaded. Consider splitting into separate panels (total vs. cause-specific mortality) for clarity.

Terminology: Ensure consistent use of "maternal mortality ratio (MMR)" vs. "maternal mortality" throughout. Clarify terms like "coexisting medical conditions" vs. "indirect causes."

Discussion Structure: Could be tightened by reducing repetition and focusing on three big global lessons (coverage + quality; system/urbanization context).

Reviewer #3: The study aimed to estimate the most contributing social determinants and health system-related factors to the decline in total and cause-specific maternal mortality in China from 2004 to 2020, by use of Bayesian kernel machine regression (BKMR) to tackle the collinearity of covariates. The data originated from Global Burden of Diseases, Injuries, and Risk Factors Study 2021 (GBD 2021), the National Health Statistics Yearbooks, and the China Statistical Yearbooks. This study is of significant importance. It will provide important experience for other regions to achieving MMR decline target in SDGs. Overall, the manuscript is well written. I just have minor comments.

1.The dependent variable in this study is time-dependent. Please explain how this issue is addressed in statistical analysis.

2.Have authors tried to analyze the contributing determinants by rural and urban areas?

---

* Please include an Author Summary after the Abstract. See:https://journals.plos.org/plosmedicine/s/revising-your-manuscript

* Please upload any figures associated with your paper as individual TIF or EPS files with 300dpi resolution at resubmission; please read our figure guidelines for more information on our requirements: http://journals.plos.org/plosmedicine/s/figures. While revising your submission, we strongly recommend that you use PLOS's NAAS tool (https://ngplosjournals.pagemajik.ai/artanalysis) to test your figure files. NAAS can convert your figure files to the TIFF file type and meet basic requirements (such as print size, resolution), or provide you with a report on issues that do not meet our requirements and that NAAS cannot fix.

After uploading your figures to PLOS's NAAS tool - https://ngplosjournals.pagemajik.ai/artanalysis, NAAS will process the files provided and display the results in the "Uploaded Files" section of the page as the processing is complete.

If the uploaded figures meet our requirements (or NAAS is able to fix the files to meet our requirements), the figure will be marked as "fixed" above. If NAAS is unable to fix the files, a red "failed" label will appear above.

When NAAS has confirmed that the figure files meet our requirements, please download the file via the download option, and include these NAAS processed figure files when submitting your revised manuscript.

* Please ensure that the study is reported according to the appropriate guideline (e.g. STROBE) and include the completed checklist as Supporting Information. When completing the checklist, please use section and paragraph numbers, rather than page numbers. Please add the following statement, or similar, to the Methods: "This study is reported as per [XXXX] guideline (S1 Checklist)."

FIGURES AND TABLES

SUPPLEMENTARY MATERIAL

REFERENCES

OBSERVATIONAL STUDIES

* Abstract: Please include the study design, population and setting, number of participants, years during which the study took place (enrollment and follow up), length of follow up, and main outcome measures.

* Please ensure that the study is reported according to the STROBE (or appropriate STOBE extension) guideline (available from: https://www.equator-network.org/reporting-guidelines/strobe) and include the completed STROBE (or STROBE extension) checklist as Supporting Information. Please add the following statement, or similar, to the Methods: "This study is reported as per the Strengthening the Reporting of Observational Studies in Epidemiology (STROBE) guideline (S1 Checklist)." When completing the checklist, please use section and paragraph numbers, rather than page numbers.

* [FOR POPULATION HEALTH/REGISTRY STUDIES] Please ensure that the study is reported according to the RECORD guideline (available from https://www.record-statement.org) and include the completed checklist as Supporting Information. Please add the following statement, or similar, to the Methods: "This study is reported as per the Reporting of Studies Conducted using Observational Routinely-Collected Data (RECORD) guideline (S1 Checklist)." When completing the checklist, please use section and paragraph numbers, rather than page numbers.

* [FOR POPULATION HEALTH ESTIMATES] Please ensure that the study is reported according to the GATHER statement (available from https://www.equator-network.org/reporting-guidelines/gather-statement) and include the completed checklist as Supporting Information. Please add the following statement, or similar, to the Methods: "This study is reported as per the Guidelines for Accurate and Transparent Health Estimates Reporting (GATHER) statement (S1 Checklist)." When completing the checklist, please use section and paragraph numbers, rather than page numbers.

* [FOR MEDIATION ANALYSES] We recommend that the study is reported according to the AGReMA statement (https://agrema-statement.org/#:~:text=AGReMA%20is%20an%20evidence%2D%20and,randomised%20trials%20and%20observational%20studies) and include the completed checklist as Supporting Information. Please add the following statement, or similar, to the Methods: "This study is reported as per the Guideline for Reporting Mediation Analyses (AGReMA) statement (S1 Checklist)." When completing the checklist, please use section and paragraph numbers, rather than page numbers.

* For all observational studies, in the manuscript text, please indicate: (1) the specific hypotheses you intended to test, (2) the analytical methods by which you planned to test them, (3) the analyses you actually performed, and (4) when reported analyses differ from those that were planned, transparent explanations for differences that affect the reliability of the study's results. If a reported analysis was performed based on an interesting but unanticipated pattern in the data, please be clear that the analysis was data driven.

* Please state in the Methods section whether the study had a prospective protocol or analysis plan. If a prospective analysis plan (from your funding proposal, IRB or other ethics committee submission, study protocol, or other planning document written before analyzing the data) was used in designing the study, please include the relevant document(s) with your revised manuscript as a Supporting Information file to be published alongside your study and cite it in the Methods section. A legend for this file should be included at the end of your manuscript. If no such document exists, please make sure that the Methods section transparently describes when analyses were planned, and when/why any data-driven changes to analyses took place. Changes in the analysis, including those made in response to peer review comments, should be identified as such in the Methods section of the paper, with rationale.

MODELLING STUDIES

The following list is derived from Geoffrey P Garnett, Simon Cousens, Timothy B Hallett, Richard Steketee, Neff Walker. Mathematical models in the evaluation of health programmes. (2011) Lancet DOI:10.1016/S0140-6736(10)61505-X:

* If pertinent, please provide a diagram that shows the model structure, including how the natural history of the disease is represented, the process and determinants of disease acquisition, and how the putative intervention could affect the system.

* Please provide a complete list of model parameters, including clear and precise descriptions of the meaning of each parameter, together with the values or ranges for each, with justification or the primary source cited and important caveats about the use of these values noted.

* Please provide a clear statement about how the model was fitted to the data, including goodness-of-fit measure, the numerical algorithm used, which parameter varied, constraints imposed on parameter values, and starting conditions.

* For uncertainty analyses, please state the sources of uncertainties quantified and not quantified [can include parameter, data, and model structure].

* Please provide sensitivity analyses to identify which parameter values are most important in the model. Uncertainty estimates seek to derive a range of credible results on the basis of an exploration of the range of reasonable parameter values. The choice of method should be presented and justified.

* Please discuss the scientific rationale for the choice of model structure and identify points where this choice could influence conclusions drawn. Please also describe the strength of the scientific basis underlying the key model assumptions.

* For studies that develop a prediction model or evaluate its performance, please ensure that the study is reported according to the TRIPOD statement (https://www.equator-network.org/reporting-guidelines/tripod-statement) and include the completed checklist as Supporting Information. Please add the following statement, or similar, to the Methods: "This study is reported as per the Transparent Reporting of a Multivariable Prediction Model for Individual Prognosis Or Diagnosis (TRIPOD) statement (S1 Checklist)." For studies using machine learning, please use the TRIPOD-AI checklist. When completing the checklist, please use section and paragraph numbers, rather than page numbers.

---

## [Decision Letter · Decision Letter 2]

7 Nov 2025

Dear Dr. Zhang,

Thank you very much for re-submitting your manuscript "Social and Health System Determinants of Maternal Mortality in More and Less Developed Regions of China: Implications for SDG 2030" (PMEDICINE-D-25-02747R2) for review by PLOS Medicine.

I have discussed the paper with my colleagues and the academic editor and it was also seen again by 2 reviewers. I am pleased to say that provided the remaining editorial and production issues are dealt with we are planning to accept the paper for publication in the journal.

The reviewers have a few remaining comments that we ask you to address in a point-by-point response. Please note in particular that we require that you to remove all causal assertions from the manuscript and frame your results as associative, as per the comments of reviewer 1. Please remove 'important' from the Abstract, qualify determinants (i.e. determinants of what) and see below for other Abstract requirements.

Please also note that we require you to respond to the comments of reviewer 2, and please refer to this reviewer's comments from the original round of review, and provide a response in full to the points originally raised.

Please ensure that the manuscript includes a definition of Eastern and Western China.

[LINK]

We look forward to receiving the revised manuscript by Nov 14 2025 11:59PM.   

Sincerely,

Alison Farrell, Ph.D.

Senior Editor 

PLOS Medicine

plosmedicine.org

Requests from Editors:

* Please confirm that your title complies with PLOS Medicine's style. Your title must be nondeclarative and not a question. It should begin with main concept if possible. "Effect of" should be used only if causality can be inferred, i.e., for an RCT. Please place the study design ("A randomized controlled trial," "A retrospective study," "A modelling study," etc.) in the subtitle (ie, after a colon).

* Please confirm that your abstract complies with our requirements, including format (three sections: Background, Methods and Findings, and Conclusions) and providing all the information relevant to this study type https://journals.plos.org/plosmedicine/s/submission-guidelines#loc-abstract

* Please ensure that the Introduction ends with a clear description of the study question or hypothesis.

* Please ensure that all abbreviations are defined at first use throughout the text.

* Please confirm that all numbers presented in the abstract are present and identical to numbers presented in the main manuscript text.

* Please review your text for claims of novelty or primacy (e.g. 'for the first time') and remove this language. In addition, please check that any use of statistical terms (such as trend or significant) are supported by the data, and if not please remove them.

* In the abstract, please include the important dependent variables that are adjusted for in the analyses.

* Please convert any stacked bar charts to another data representation for example a table, or other type of graph. If that is not possible, please explain why and provide a Table containing the data in the bar chart (e.g. in Fig. 2).

In Fig. 3, please lighten the dark blue colour (Hemorrhage) for improved clarity.

* Where data points are discrete, please ensure that they are depicted in the figures as discrete data and not as a continuous line, e.g. in Fig. 1.

* Please provide the unadjusted comparisons as well as the adjusted comparisons in all relevant Tables

* Please specify the variables controlled for in all relevant Tables

Comments from Reviewers:

Reviewer #1: The authors have comprehensively addressed many of my original concerns and those of other reviewers, particularly regarding some of the additional analyses they present. In general, I am satisfied with these revisions, but did still have two lingering comments - one that I would like to see addressed, and another that might serve merely as a suggestion.

Most importantly, the language used in the model still inappropriately makes causal assertions. for instance, the word "determinant" implies that these social factors are determinants of mortality or health outcome, which is a causal association. Furthermore, the word "important" is being used as a proxy for magnitude of effect, which might not necessarily connote "importance" in terms of the concept we regularly think of. Some of the suggested language is indeed appropriate (e.g., directly stating that something was "associated with a higher value of the exposure-response relationship,"), and so I encourage the authors to use such language throughout, instead of anything that may - incorrectly or not - be construed as implying a causal relationship (e.g., "important determinant for").

As a potential suggestion, I wonder whether the current "sensitivity analyses" might actually serve better as the main analysis. For instance, the mixed model that directly addresses the question "an x increase in y is associated with a z change in w, while adjusting for the other abc..." is directly interpretable and directly addresses research questions of interest regarding the magnitude of relationships. The sensitivity analysis might then be the BKMR, which answers the potential challenge to the analysis of "we know some of these are correlated - what might "get chosen" if we only had to choose a subset of these individual predictors."

Regardless, I am satisfied with the response to original review and the revised manuscript. The only real comment I have is on the lingering use of causal language.

Reviewer #2: The authors of the manuscript has addressed most of my comments in the revised version. However, they failed to address one major comment, that is, the quality of data derived from the national health statistics published in the 2010s. The answer to my comment (No 2) was vague and did not show their understanding of potential implications for the use of such poor quality data. I understand that it is a very tough question that may not be easily addressed. Should the authors not be able to address this issue, they should at least put it as one of main limitations of the study.

[LINK]

---

## [Editor Report · Decision Letter 3]

14 Nov 2025

Dear Dr Zhang, 

On behalf of my colleagues and the Academic Editor, Margaret Kruk, I am pleased to inform you that we have agreed to publish your manuscript "Social and health system factors associated with maternal mortality in more and less developed regions of China: Population health estimates using provincial-level data" (PMEDICINE-D-25-02747R3) in PLOS Medicine.

Please also address these editorial requests:

Title: replace "more and less developed regions of" with "Eastern and Western" so that the title does not age as development changes occur in China

Line 36: capitalize Eastern and Western

Line 56: delete 'the remarkable progress' and replace with 'improvements'

Line 59: revise "it is vital" to "it may be vital"

Line 337: delete 'and' before 'an'

Add the URLs for funders websites to the funding information.

Please check capitalization of Eastern and Western throughout (should be capitalized when next to China).

PRESS

Sincerely, 

Alison Farrell, Ph.D. 

Senior Editor 

PLOS Medicine